# Pulp obtained after isolation of starch from red and purple potatoes (*Solanum tuberosum L.*) as an innovative ingredient in the production of gluten-free bread

**Dorota Gumul**[1]*, **Jarosław Korus**[1], **Magdalena Surma**[2], **Rafał Ziobro**[1]

**1** Department of Carbohydrate Technology, Faculty of Food Technology, University of Agriculture in Krakow, Krakow, Poland, **2** Department of Plant Products Technology and Nutrition Hygiene, Faculty of Food Technology, University of Agriculture in Krakow, Krakow, Poland

* rrgumul@cyf-kr.edu.pl

**Data Availability Statement:** All relevant data are within the manuscript and its Supporting Information files.

**Funding:** The research was financed by the Ministry of Science and Higher Education of the

## Abstract

Starch based gluten-free bread (formulations containing mixture of corn and potato starch with hydrocolloids) are deficient in nutrients and do not contain health promoting compounds. Therefore they could be supplemented with raw materials rich in such components, especially antioxidants. Among them pseudo-cereals, seeds, fruits and vegetables are often applied to this purpose. Potato pulp produced by processing red fleshed (Magenta Love) and purple fleshed (Violetta) varieties could become a new innovative substrate for gluten-free bread enrichment, because of high levels of endogenous polyphenols, namely flavonoids, flavonols, phenolic acids and especially anthocyanins with high antioxidant potential, as well as dietary fiber. Study material consisted of gluten-free bread enriched in the pulp. Dietary fiber, acrylamide content and antioxidant and antiradical potential of the bread were determined. Sensory evaluation included crumb elasticity, porosity and other characteristics, taste and smell. Among all analyzed gluten-free breads, the sample containing 7.5% share of freeze-dried red potato pulp Magenta Love was characterized by high content of phenolic compounds and dietary fiber, pronounced antioxidant activity, low levels of potentially dangerous acrylamide and good physical and sensory characteristics. Therefore such an addition (7.5% Magenta Love) could be recommended for industrial production of gluten-free bread.

## Introduction

During laboratory or industrial isolation of potato starch all other components of plant tissue have to be separated, into dissolved and insoluble fractions, which are called potato-juice and potato pulp. Potato-juice is one of the main streams of organic by-products from potato-starch processing. Production of 1 ton of potato starch results in approximately 3.5 tons of extracted potato-juice as by-product [1]. Potato-juice contains mainly valuable protein, carbohydrates and minerals. Because of the high quantity of protein, it is mainly used for the production of

Republic of Poland. The funders had no role in study design, data collection and analysis, decision to publish, or preparation of the manuscript.

**Competing interests:** The authors have declared that no competing interests exist.

coagulated protein. The preparation usually contains approximately 80% of protein, characterized by 90% digestibility, which is mainly used as feed, but could also be used for food processing industry. Additionally potato-juice as an organic by-product reach in protein and carbohydrates could be used for the production of biogas through anaerobic digestion [2].

Potato pulp, the second by-product formed during the manufacture of potato starch, is also produced in large quantities, even exceeding the above mentioned numbers for potato-juice. The quantity of wet potato pulp corresponding to the manufacture of 1 ton of starch is around 4.5-5 tons [3]. Potato pulp contains large quantities of fiber (cellulose and hemicellulose, pectin) and other insoluble components of plant cells. The most prominent uses of potato pulp involve its application as animal feed and production of dietary fiber on industrial scale in Europe [4–7]. Moreover potato pulp is used as a medium for bioethanol production, often accompanied by enzymatic hydrolysis of cellulose. Unfortunately cellulose chains are strongly recalcitrant to enzymatic treatment, which significantly diminishes the utilization of this by-product [4, 5]. Another limitation is due to pulp's microbiological instability caused by high water content, which can be effectively eliminated by drying, but requires high energy input. In addition to the above mentioned utilization possibilities, new, additional ways of its use must be found, that may at least partially omit the above mentioned limitations. Special attention should be paid to its complex chemical composition. Potato pulp is a heterogeneous mixture of many components, in which fiber is accompanied by proteins, little amount of fat, minerals and nitrogen-free extract Attention should be given to protein of high biological value, which contains high levels of essential amino acids, and therefore is complete (similarly to egg white and animal proteins), and lipids with a significant proportion of polyunsaturated acids (linoleic and linolenic acids) [8–10]. Nevertheless the largest part of potato pulp is composed of fiber fraction. Although it has already been pointed out that pulp is a source of dietary fiber [6, 7], this issue has not been associated with polyphenols contained in pulp. It should then be emphasized that an integral part of the fiber in potato, which consists mainly of cellulose, hemicellulose, lignin and pectin [5] encompasses bioactive compounds from the group of polyphenols. Considering the above mentioned composition we could regard potato pulp as a concentrate of pro-health constituents, especially if red and purple potatoes are used for starch isolation. In the latter case the levels of phenolic compounds are 2 – 3 times higher in comparison to light flesh varieties [11, 12], which makes them a rich source of these bioactive compounds. Red and purple potatoes contain large quantities of phenolic acids (chlorogenic, neochlorogenic and cryptochlorogenic, gallic, ferulic, protocatechic, caffeic, cynnaminic, synapic and p-coumaric acid), flavonoids (catechin, epicatechin, kempferol, naringenin, rutin), and especially heat stable anthocyanins, acylated with phenolic acids (mainly ferulic and p-coumaric) [11, 13] and reveal antiinflammatory, antiviral and anticancer properties [14]. This type of potato pulp isolated from potatoes with red and purple flesh may be considered as a cheap and valuable concentrate of pro-health compounds, appropriate for food uses as an alternative to existing additives used in the production of gluten-free bread, such as pseudocereals, linseed, and fiber preparations [15]. The need for such alternatives is obvious in the context of increasing demand for gluten-free products.

Celiac disease is a genetic enteropathy, with an autoimmune basis, which leads to a disappearance of intestinal villi, which in turn impedes absorption of nutrients (fiber, minerals, protein) from food. The deficiencies caused by malabsorption negatively influence immunological status, promote the formation of osteoporosis, rickets, anemia, and slow down metal and physical development [16]. It has been suggested that a close relation exists between gluten and development of the above mentioned diseases, which requires strict adherence to gluten-free diet by the people affected by celiac diseases. Disorders induced by gluten proteins also include gluten ataxia, wheat allergies, and gluten sensitivity [16, 17], which require elimination of

gluten based products, such as wheat bread. Unfortunately, gluten-free bread is characterized by much lower nutritional value than wheat bread and lack of health promoting substances, which contribute to many diseases in people with celiac disease, such as osteoporosis, esophageal cancer, infertility, etc. [17]. The potato pulp derived from potatoes with red and purple flesh with a high content of pro-health components may become a perfect innovative addition to gluten-free products, which will cause a large increase in phenolic compounds and fiber in this type of products. It may be suggested that the introduction of potato pulp, containing high levels of fiber, responsible for proper functioning of dietary tract, significant amounts of polyphenols with chemopreventive activity, and complete protein together with minerals, which according to Yang *et al.* [3] represent 4 and 1.5% of its dry substance, respectively, will result in an enrichment of gluten-free bread in the above mentioned components. Such enrichment is important, especially for starch based bread, the recipes of which consist mainly of corn and potato starch. Such gluten-free breads are very poor in nutrients such as proteins and minerals compared to breads made from corn and rice flour. These types of breads do not contain any health promoting substances (fiber and polyphenols) which exert chemopreventive role in human nutrition. The addition of pulp obtained from red and purple potatoes to gluten-free breads will also allow to extend the range of gluten-free products and thus show an innovative way of managing this by-product. It therefore seems fully justified to undertake studies in this area.

It should also be stressed that in the context of growing consumer awareness of food composition, it is equally important to have a high pro-health value and a small amount of harmful substances, such as acrylamide. Acrylamide is a compound formed from free amino acid, mainly asparagine and reducing sugars (glucose fructose, maltose) during thermal operations (baking, frying) and is considered to be a carcinogenic compound [18–20]. That is why it is so important to consider not only the pro-health value of gluten-free bread with potato pulp but also the content of acrylamide in it.

Therefore, the aim of the study was to analyze the influence of different levels of freeze-dried pulp from red and purple potatoes on the content of polyphenols and fiber in gluten-free breads. Moreover, their antioxidant activity was estimated, as well as the determination of acrylamide was accomplished in the analyzed gluten-free breads with the above mentioned potato pulp.

## Materials and methods

### Materials

Study material consisted of gluten-free bread with a share of 5; 7.5 and 10% freeze-dried potato pulp obtained after laboratory extraction of starch from potatoes with red (Magenta Love) and purple (Violetta) flesh.

Potato pulp was obtained as a by-product in laboratory isolation of potato starch, according to Wischmann *et al.* [21], subsequently freeze-dried, and included in bread formulations.

### Methods

**Bread preparation.** Control bread was baked according to the following recipe: corn starch 552g, potato starch 138g, freeze-dried yeast 34.5g, guar gum 11.5g, pectin 11.5g, sucrose 13.8g, salt 13.8g, canola oil 21g, water 655g. In other samples part of potato and corn starch (5, 7.5, 10%) was replaced with appropriate freeze-dried potato pulp. All ingredients were mixed for 5 minutes (Laboratory Spiral Mixer SP 12, Diosna, Germany). The dough was fermented for 15 min at 35 ˚C and relative moisture level 80%. After initial proofing the dough was divided into 250 g pieces into greased baking pans and fermented for another 20 min under

the above mentioned conditions. Bread was baked at 230 ˚C for 30 min. in electric oven MIWE Condo type CO 2 0608 (MIWE GmbH, Germany). The loaves were removed from pans and cooled to ambient temperature. Each formulation was baked in two independent batches, 5 loaves in each batch. The whole loaf, after being cut into 1 cm thick slices and air-dried, was ground, sifted through a 1 mm$^2$ mesh screen. The sample was thoroughly mixed before being taken for chemical analyses.

The following analyses were performed for sample:

Chemical composition – content of protein, fat, ash – for pulp obtained from red and purple potatoes was determined according to AOAC 2006 [22]. The measurements were done in duplicate.

Antioxidant constituents and antiradical activity were determined in the ethanol extracts. 0.6 g of the sample was dissolved in 30 cm$^3$ 80% ethanol, shaken in a darkness for 120 minutes (electric shaker: type WB22, Memmert, Schwabach, Germany), and centrifuged (15 min., $1050 \times g$).) in centrifuge (type MPW-350, MPW MED. Instruments, Warsaw, Poland). The supernatant was decanted and stored at -20ºC for further analyses [23].

Determination of total polyphenols content (TPC) was done by two spectrophotometric methods a) using Folin-Ciocalteu reagent (with F-C reagent), according to Singleton *et*. *al*. [24] and b) without using Folin-Ciocalteu reagent (without F-C reagent), according to Mazza *et*. *al*. [25], with the modification of Oomah *et*. *al*. [26]. The content of phenolic acids was measured using a spectrophotometrical method, according to Mazza *et al*. [25], with the modification of Oomah *et al*. [26]. The content of flavonols was measured spectrophotometrically, according to Mazza *et al*. [25], with the modification of Oomah *et al*. [26]. Determination of the content of anthocyanins was done spectrophotometrically, according to Mazza *et al*. [25], with the modification of Oomah *et al*. [26]. The content of flavonoids was evaluated using a spectrophotometrical method, according to El Hariri *et*. *al*. [27]. The measurements were done in four replications.

Additionally antiradical activity was assessed using analytical methods, namely ABTS [28]. Results of antiradical activity were expressed as TEAC (Trolox Equivalent Antioxidant Capacity - mg Trolox/ g dry mass of sample). The measurements were done in duplicate.

Content of non-starch polysaccharides, i.e. total, soluble and insoluble dietary fiber, by the method 32-07 of AACC [29]. The measurements were done in four replications.

Moreover the content of acrylamide (AA) has been determined. For sample preparation according to Surma *et*. *al*. [30] modified QuEChERS method was applied. AA qualitative and quantitative analysis were done using HPLC-UV/Vis (according to methods Matrconi *et al*., [31]). The measurements were done in four replications.

**Sensory analysis of bread.** The panel of 15 trained persons assessed the bread according to the Polish Standard (PN-A-74108:1996) [32] in terms of external appearance (maximum 5 points), crust color (maximum 3 points), crust thickness (maximum 4 points) and other characteristics (maximum 4 points), crumb elasticity (maximum 4 points), crumb porosity (maximum 3 points) and other characteristics (maximum 3 points) as well as taste and smell (maximum 6 points). The number of points in the evaluation of each trait reflected its weight and influence on the quality of bread, and the evaluation was made by comparing the individual traits with their description in the evaluation table included in the standard.

## Statistical analysis

The experimental data were subjected to analysis of variance (Duncan's test), at the confidence level of 0.05, by the use of software Statistica v. 8.0 (Statsoft, Inc., Tulsa, OK., USA). The Pearson correlation coefficients between selected parameters were also calculated.

## Results and discussion

### Characteristics of potato pulp obtained from red and purple potatoes

Potato pulp obtained from red and purple potatoes contained protein and ash at 6.44 (g·100g$^{-1}$ d.m.) and 2.37 (g·100g$^{-1}$ d.m.) (pp ML) and 5.98 (g·100g$^{-1}$ d.m.) and 2.05 (g·100g$^{-1}$ d.m.) (pp V) respectively. In contrast, in the study on potato pulp from yellow fleshed potatoes the amount of proteins was at the level of 5.1-6.5 (g·100g$^{-1}$ d.m.) and ash 2.7-3.3 (g·100g$^{-1}$ d.m.), crude fiber was 20.5 (g·100g$^{-1}$ d.m.) [7, 33, 34]. Thus, it can be concluded that in terms of the amount of protein the pulp from potato red/purple compared to yellow potato pulp did not differ. In the case of ash, its amount in pulp from yellow potatoes was 30% higher than that of red or purple.

As mentioned earlier, potato pulp is a heterogenous mixture of many components, mainly classified as dietary fiber and less other components such as protein, ash, fat and nitrogen-free extract. Taking into account the fact that polyphenols are an integral part of fiber and that the content of polyphenols in red/purple potatoes is 2-3 times higher than in yellow ones [11, 12], the same trend in polyphenol content in the pulps of these two types of potatoes can be suggested. It is therefore important to analyze the pulp of red/purple potatoes as a source of polyphenols due to the lack of research on the subject. At the same time, it should be mentioned that there are no studies on the analysis of polyphenols in the pulp of yellow potatoes. This proves the innovativeness of the conducted research, as until now scientists have only dealt with potato pulp as a source of fiber, completely ignoring the fact that polyphenols are an integral part of this fiber. Therefore, the potato pulp obtained from red and purple potatoes is discussed below as a source of both fiber and polyphenols.

Table 1 represents total dietary fiber and its soluble and insoluble fractions, total polyphenols, flavonoids, flavonols, anthocyanins and phenolic acids, together with antiradical activity of freeze-dried samples of potato pulp obtained from potatoes with red flesh (Magenta Love) and purple flesh (Violetta).

It could be observed that the pulp left after isolation of starch from red potatoes (Magenta Love) contained 23% more insoluble fiber and 17% less soluble fiber in comparison to the pulp from purple potatoes (Violetta). Total dietary fiber content was larger in the case of Magenta Love (14% more than in Violetta). Taking into account that total dietary fiber in freeze-dried red and purple potatoes ranged between 7 and 8.76 g/100 g, its soluble fraction varied between 1.87 – 2.74 and insoluble fraction 3.9 – 6 g/100 g [35] it could be concluded that the pulp is much more abundant in fiber (Table 1) than respective freeze-dried tubers [35]. In the case of total dietary fiber (TDF) the content in pulp is twice as high as in freeze-dried potato tissue, in

**Table 1. Characteristics of potato pulp obtained from red and purple potatoes.**

| Kind of potato pulp | Insoluble dietary fiber (g·100g$^{-1}$ d.m) | Soluble dietary fiber (g·100g$^{-1}$ d.m) | Total dietary fiber (g·100g$^{-1}$ d.m) | TPC (with F-C reagent) (mg catechin ·100g$^{-1}$ d.m.) | TEAC (mg Trolox· g$^{-1}$ d.m.) |
|---|---|---|---|---|---|
| ppML | 16.47±0.21 [b*] | 3.03±0.06 [a] | 19.5±0.05 [b] | 359 ±0.1 [b] | 64.97±2.8 [b] |
| ppV | 13.4±0.11 [a] | 3.65±0.13 [b] | 17.05±0.12 [a] | 335±0 [a] | 54.65±1.38 [a] |
| Kind of potato pulp | TPC (without F-C reagent) (mg catechin ·100g$^{-1}$ d.m.) | Phenolic acids (mg ferulic acid·100g$^{-1}$ d.m.) | Flavonoids (mg rutin· 100 g$^{-1}$ d.m.) | Flavonols (mg qercetin·100 g$^{-1}$ d.m.) | Anthocyanins (mg cyanidin-3-glucoside·100 g$^{-1}$ d.m.) |
| ppML | 345.03±0.34 [b] | 66±1 [b] | 134±1.7 [b] | 39.73±2.9 [b] | 104.03±0 [b] |
| ppV | 313.41±0 [a] | 52±2 [a] | 114±2.42 [a] | 29.87±0 [a] | 97.05±1.65 [a] |

ppMl - potato pulp obtained from red potatoes variety Magenta Love

ppV - potato pulp obtained from purple potatoes variety Violetta

* Presented data are mean values ± standard deviation (values signed the same letters in particular columns are not significant at 0.05 level of confidence

the case of insoluble dietary fiber (IDF) and soluble dietary fiber (SDF) the increase is 3-4-fold and 1.5-fold, respectively. The increase in fiber concentration is very important because of physiological role of fiber fraction on human organism. Insoluble dietary fiber is especially needed for prevention and treatment of malfunctions in large bowel (habitual constipation, irritable bowel syndrome, hemorrhoids and colorectal diverticulosis). On the other hand the soluble fraction of dietary fiber has hypocholesterolemic, hypoglycemic and anti-cancerogenic effects [36, 37]. Potato pulp is a natural concentrate of endogenous substances of dietary fiber, containing non starch polysaccharides of potato cell walls, mainly hemicellulose, lignin and pectin. These polysaccharides are bound via hydrogen and ionic bonds, as well as hydrophobic interactions with phenolic compounds present in potatoes. Especially important seem to be the interactions between anthocyanins and pectins, although their mechanism is still unclear [38]. It could be stated that polyphenols are an integral part of fiber present in potato pulp, which has not yet been generally recognized by food scientists and producers. Considering potato pulp as a natural concentrate of endogenous phenolic compounds bound in fiber fraction may offer a new innovative possibility of using this by-product. The total content of polyphenols was significant in the analyzed samples (Table 1), while the content of polyphenols was higher in the samples originated from red potatoes Magenta Love (Table 1), than purple ones Violetta, irrespective of the applied determination method (with or without Folin-Ciocalteau (F-C) reagent). It should be remembered that Folin-Ciocalteau reagent gives a positive result not only with polyphenols but also amino acids, proteins, saccharides, therefore apart from popular method of Singleton *et al.* [24] based on the above mentioned reagent, another assessment was applied without the use of Folin-Ciocalteau reagent (Mazza *et al.*, [25] with modification of Oomah *et al.*, [26]). In the case of other phenolic compounds, namely flavonoids, flavonols, phenolic acids and anthocyanins, their elevated quantities were also measured in the pulp from Magenta Love potatoes, in comparison to Violetta: by 17%, 33%, 27% and 6.6%, respectively. It was observed that higher amount of phenolic compounds in the pulp from Magenta Love potatoes cause higher their antiradical activity than activity of pulp from Violetta potatoes (Table 1). Confirmed high levels of fiber and phenolic compounds in the analyzed samples of potato pulp, provides opportunity to use them as a natural concentrate of endogenous substances with pro-health activity for the production of food, such as gluten-free bread.

## The influence of potato pulp obtained from red and purple potatoes on the content of pro-health constituents in gluten-free bread

Gluten-free bread as already mentioned is poor in nutrients and above all health promoting constituents, therefore the introduction of pulp from red and purple potatoes, which is a source of health promoting compounds (polyphenols and fiber) is highly justified. In the case of dietary fiber and its soluble/insoluble fractions the content of these constituents were larger in the case of gluten-free breads with a share of freeze-dried pulp derived from red and purple potatoes by 24, 19 and 28% in comparison to control. It was observed that the rise of these compounds was proportional to the addition of pulp, irrespective of its origin. The influence of potato variety in this case was not statistically significant (Table 2).

Taking into account total polyphenol content (TPC) it could be observed that the introduction of 5 to 10% share of freeze-dried pulp originating from red and purple potatoes caused the increase in comparison to control between 7.7 to 197%, when Folin-Ciocalteau reagent was used, and 4 to 7 times if the method without this reagent was applied (Tables 2 and 3). It could be noticed, that irrespective of the addition level the application of the pulp Magenta Love in gluten-free bread formulation was more efficient than the use of pulp Violetta, due to

**Table 2. Dietary fiber and Total Phenolic Content (TPC) and antioxidant activity of gluten- free breads with potato pulp obtained from red and purple potatoes.**

| Kind of breads | Insoluble dietary fiber (g·100g$^{-1}$ d.m.) | Soluble dietary fiber (g·100g$^{-1}$ d.m.) | Total dietary fiber (g·100g$^{-1}$ d.m) | TPC (with F-C reagent) (mg catechin·100g$^{-1}$ d.m.) | TEAC (mg Trolox·g$^{-1}$ d.m.) |
|---|---|---|---|---|---|
| **Control** | 2.7±0.1 [a*] | 2.03±0 [a] | 4.73±0.32 [a] | 13± 2 [a] | 3.08±1.22 [a] |
| **GFB +5ppML** | 3.26±0.02 [b] | 2.16±0.1 [b] | 5.42±0.13 [b] | 18.4±1.14 [b] | 11.7±0.7 [c] |
| **GFB +5ppV** | 3.17±0.07 [b] | 2.16±0 [b] | 5.33±0.18 [b] | 14.01±4.02 [b] | 9.5±0.55 [b] |
| **GFB +7.5ppML** | 3.34±0.03 [c] | 2.65±0,09 [c] | 5.99±0.07 [c] | 27.8±0 [d] | 19.83±1.67 [e] |
| **GFB +7.5ppV** | 3.48±0.1 [c] | 2.46±0.11 [c] | 5.94±0.02 [c] | 23.9±0.87 [c] | 13.44±0 [d] |
| **GFB +10ppML** | 3.54±0.07 [cd] | 2.60±0.12 [c] | 6.14 ±0.15 [d] | 38.71±1.05 [e] | 39.4±1.79 [g] |
| **GFB +10ppV** | 3.91±0.41 [d] | 2.51±0.18 [c] | 6.41±0.18 [d] | 27.47± 1 [d] | 33.02±0.91 [f] |

Control - control gluten-free bread, GFB +5ppML - gluten-free bread with 5% share of pulp potatoes obtained from red potatoes variety Magenta Love, GFB +5ppV - gluten-free bread with a 5% share of pulp potatoes obtained from purple potatoes variety Violetta, analogical the other abbreviations

* Presented data are mean values ± standard deviation (values signed the same letters in particular columns are not significant at 0.05 level of confidence)

much higher content of the bioactive compounds in respective pulp potato varieties (Tables 1–3). In the case of phenolic acids, their presence in both types of pulp potato (Table 1), was not accompanied by their detection in respective bread (Table 3). It is in agreement with earlier findings of Maillard and Bersett [39] who noticed thermal decarboxylation of these compounds e.g to 4-vinyloguaiacol. On the other hand significant changes could be observed for the next group of phenolic compounds – flavonoids. Their increase in gluten-free bread after introduction of potato pulp was significant and ranged between 15 and 28.5 times in the case of Magenta Love and 13 – 22 times in the case of Violetta, as compared to control. It was also found that the increase in flavonoids was proportional to the amounts of the applied pulp (Table 3). In the case of two subgroups of flavonoids – flavonols and anthocyanins, their

**Table 3. Phenolic compounds in gluten-free breads with potato pulp obtained from red and purple potatoes.**

| Kind of breads | TPC (without F-C reagent) (mg catechin·100g$^{-1}$ d.m.) | Phenolic acids (mg ferulic acid·100g$^{-1}$ d.m.) | Flavonoids (mg rutin· 100 g$^{-1}$ d.m.) | Flavonols (mg qercetin·100 g$^{-1}$ d.m.) | Anthocy-anins (mg cyanidin-3-glucoside·100 g$^{-1}$ d.m.) |
|---|---|---|---|---|---|
| **Control** | 1.8±0.12 [a*] | nd | 1.93±0.87 [a] | nd | nd |
| **GFB +5ppML** | 10.14±0.23 [d] | nd | 28.74±1.02 [c] | 2.31±0.28 [a] | 7.88±0.27 [b] |
| **GFB +5ppV** | 8.92±0 [b] | nd | 25±1.15 [b] | 1.86±0.25 [a] | 5.78±0.23 [a] |
| **GFB +7.5ppML** | 11.43±0.77 [d] | nd | 43.61±0 [e] | 3.43±0.2 [c] | 8.65±0.34 [c] |
| **GFB +7.5ppV** | 9.75±0 [c] | nd | 30.03±1.13 [c] | 2.75±0 [b] | 7.21±0.53 [b] |
| **GFB +10ppML** | 15.37± 0.95 [e] | nd | 55.6±0.3 [f] | 5.71±0.23 [d] | 9.23± 0 [d] |
| **GFB +10ppV** | 11.29±0.23 [d] | nd | 41.82±0 [d] | 3.67±0.17 [c] | 8.34±0.13 [c] |

nd - not destinated

Control- control gluten-free bread, GFB +5ppML - gluten-free bread with 5% share of pulp potatoes obtained from red potatoes variety Magenta Love, GFB +5ppV - gluten-free bread with a 5% share of pulp potatoes obtained from purple potatoes variety Violetta, analogical the other abbreviations

* Presented data are mean values ± standard deviation (values signed the same letters in particular columns are not significant at 0.05 level of confidence)

presence in control was not detected, but significant quantities could be found after incorporation of potato pulp. Pulp from Magenta Love has contributed to a much higher content of flavonoids, flavonols and anthocyanins in gluten-free breads than Violetta pulp. This should be explained by the fact that the higher content of these, mentioned above, pro-health ingredients was in the pulp itself (Tables 1 and 3).

Despite many authors [15, 40] state, that baking causes a decrease in polyphenols (approx 60%) because of their thermal, enzymatic and oxidative degradation, and some other processes involving oxidation, isomerization and complex formation with some other food constituents, including polysaccharides [41] the applied addition of potato pulp isolated from red and purple potatoes resulted in the presence of large quantities of bioactive compounds from polyphenol group in gluten-free bread. This seems to be highly important from dietetic point of view. High efficiency of red and purple potatoes in enrichment of bioactive compounds is evident if we compare the absence of anthocyanins in control gluten-free bread with significant level of these components after the introduction of freeze-dried potato pulp. It is especially important taking into account anti-inflammatory, anti-viral and anti-bacterial properties of anthocyanins [14]. Moreover their presence in the diet reduces the risk of carcinogenic and coronary diseases and protects against Alzheimer's disease and diabetes [11, 42].

Analyzing antiradical activity it could be observed that gluten-free bread with freeze-dried red potato pulp Magenta Love revealed 4 to 13 times higher antioxidant activity than control. Analogical increase in the case of purple potato pulp Violetta ranged from 3 to 11 times. High antiradical activity of gluten-free bread produced with a share of pulp originating from color potatoes corresponds strictly to the presence of TPC, flavonoids and anthocyanins, which could be seen in highly positive correlations between ABTS on one hand and TPC, flavonoids and anthocyanins on the other, which equaled: 0.92; 0.89 and 0.72, respectively.

Concluding – the presence of freeze-dried red potato pulp Magenta Love at the level 10% in gluten-free bread was the most beneficial, as it supplied the highest amounts of all analyzed phenolic compounds and provided high antioxidant potential to the product, which was additionally accompanied by significant amounts of dietary fiber and its soluble and insoluble fractions (Tables 2 and 3).

## Acrylamide content

In the case of 5% share of freeze-dried red potato pulp Magenta Love in gluten-free bread no statistically significant decrease of acrylamide could be observed in comparison to control sample (gluten-free bread baked without the addition of potato pulp) (Table 4). When the level of addition was 7.5% or 10% a decrease in acrylamide was statistically significant and represented 13 and 8% of the initial value, respectively – though the differences between 7.5% and 10% share of the pulp was not statistically significant. In the case of pulp derived from purple potato variety Violetta, the application of 5% and 10% addition in gluten-free bread formulation caused a statistically significant decrease in acrylamide level as compared to control, while no statistically significant difference in comparison to control could be seen when 7.5% share of the pulp was applied (Table 4). A decrease in acrylamide level could be due to the presence of substantial amounts of flavonoids, which exhibit antioxidant potential and were recently found to hamper Maillard reaction due to the carbonyl-trapping capacity of these compounds [43]. The differences in observed tendencies of acrylamide formation between pulp derived from red and purple potatoes are probably due to the changes in composition of individual anthocyanins.

However, there is no limits for acrylamide in food, according to Commission Regulation (EU) 2017/2158 [44] benchmark levels for the presence of acrylamide in foodstuffs have been established. Food was classified in 10 categories, one of which is fresh bread. We can

**Table 4. Acrylamide content in gluten-free breads with potato pulp obtained from red and purple potatoes.**

| Kind of breads | Acrylamide content ($\mu g \cdot kg^{-1}$) |
|---|---|
| Control | 849.5±34.8 [c*] |
| GFB +5ppML | 838.6± 20.6 [bc] |
| GFB +5ppV | 787.0± 22.5 [ab] |
| GFB +7.5ppML | 743.6 ± 8.7 [a] |
| GFB +7.5ppV | 820.0 ± 46.1 [bc] |
| GFB +10ppML | 782.4 ± 35.1 [a] |
| GFB +10ppV | 785.6 ± 14.3 [a] |

Control- control gluten-free bread, GFB +5ppML - gluten-free bread with 5% share of pulp potatoes obtained from red potatoes variety Magenta Love, GFB +5ppV - gluten-free bread with a 5% share of pulp potatoes obtained from purple potatoes variety Violetta, analogical the other abbreviations

* Presented data are mean values ± standard deviation (values signed the same letters in particular columns are not significant at 0.05 level of confidence)

distinguish two subcategories in it: wheat based bread and soft bread other than wheat based bread for which benchmark level for the presence of acrylamide have been set at 50 and 100 $\mu g \cdot kg^{-1}$, respectively. Taking into account the content of acrylamide in the studied bread with or without potato pulp, the detected levels of this component were larger than benchmark level. Such a high content of acrylamide may be due to the elevated levels of reducing sugar, which according to Bråthen and Knutsen [45] are more important for acrylamide formation than the level of asparagine. Reducing sugars could be formed by microbial or thermal degradation of corn and potato starch which are included in bread formulations. Also in the study of Crawford *et. al.* [46] concerning gluten-free commercial flatbreads based on tapioca starch and potato starch the levels of acrylamide were high, reaching 1880-2070 μg kg-1. It should be stressed however, that the introduction of freeze-dried red and purple potato pulp into bread formulations resulted in a decrease of acrylamide in final products, which also seems an important result of this study.

## Sensory assessment of bread

Crust and crumb color was significantly affected by the choice of applied pulp, and received lower scores in the case of purple variety Violetta in comparison to red one Magenta Love, irrespective of the applied level. On the other hand the changes between control and samples (GFB+5ppML GFB+7.5ppML) were not statistically significant. Smaller variation between scores received by gluten-free bread containing similar levels of potato pulp were observed for crust thickness and other properties.

Crumb elasticity was negatively affected by the addition of both applied pulp sources only at the highest applied level in comparison to control. In the case of crumb porosity all the loaves containing potato pulp derived from purple potatoes Violetta and the bread containing 10% of red potato pulp Magenta Love were scored significantly lower in comparison to control. Similar trend could be seen for other crumb characteristics, smell and taste, as well as overall appearance (Fig 1A–1C).

In general, sensory assessment of gluten-free bread supplemented with freeze-dried red potato pulp Magenta Love were better evaluated in comparison to those containing purple potato pulp Violetta. Additionally, together with an increasing share of added freeze-dried pulp the sensory scores were decreasing. The lowest values were obtained by bread containing 10% addition of potato pulp, irrespective of its origin.

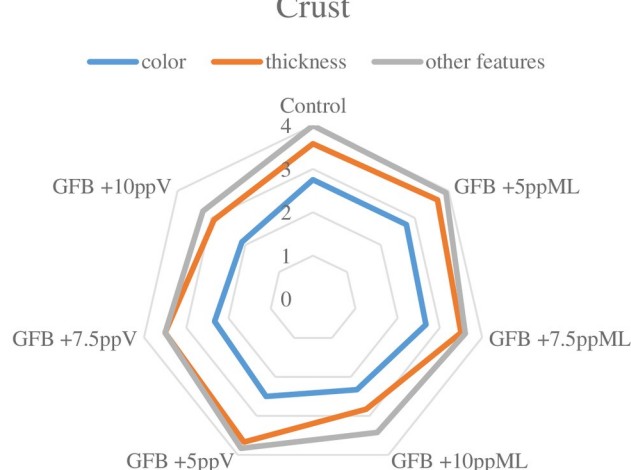

A

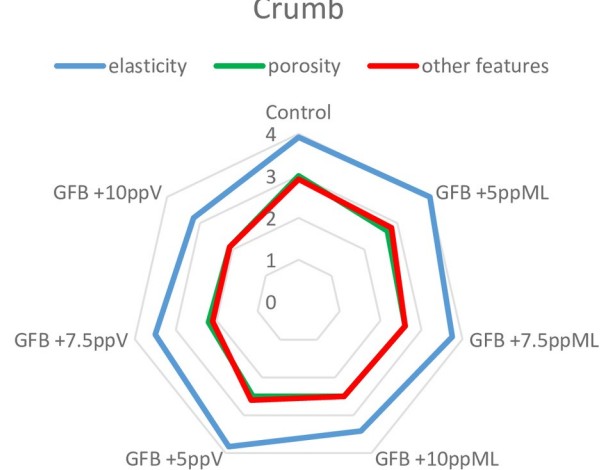

B

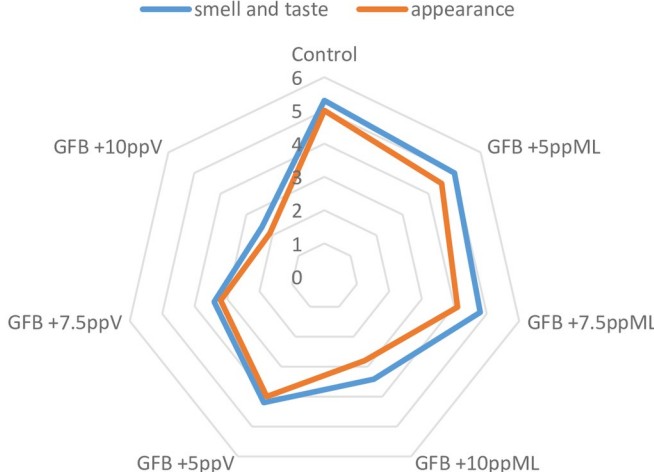

C

**Fig 1.** Sensory analysis of crust (A), crumb (B), smell, taste and appearance (C) properties of gluten free bread enriched with potato pulp.

## Conclusions

It was confirmed that pulp isolated from color flesh potatoes is a rich source of pro-health compounds, namely fiber and polyphenols. It was also found that it could be effectively used to enrich gluten-free bread in phenolic compounds and fiber, not deteriorating their physical properties and sensory scores, with the exception of the maximum applied level - 10%.

Among all analyzed gluten-free breads, the samples containing 7.5% share of freeze-dried red potato pulp Magenta Love was characterized by high content of phenolic compounds and dietary fiber, pronounced antioxidant activity, low levels of potentially dangerous acrylamide and good physical and sensory characteristics. Therefore such an addition (7.5% Magenta Love) could be recommended for industrial production of gluten-free bread.

## Author Contributions

**Conceptualization:** Dorota Gumul.

**Formal analysis:** Dorota Gumul, Jarosław Korus, Magdalena Surma, Rafał Ziobro.

**Investigation:** Dorota Gumul.

**Supervision:** Dorota Gumul.

**Writing – review & editing:** Dorota Gumul.

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
