## [Decision Letter · Decision Letter 0]

31 Dec 2019

PONE-D-19-31585

Pulp obtained after isolation of starch from red 1 and purple potatoes (Solanum tuberosum L.) as an innovative ingredient in the production of gluten free bread

PLOS ONE

Dear prof Gumul,

Thank you for submitting your manuscript to PLOS ONE. After careful consideration, we feel that it has merit but does not fully meet PLOS ONE’s publication criteria as it currently stands. Therefore, we invite you to submit a revised version of the manuscript that addresses the points raised during the review process.

Major revisions are necessary to improve the paper.  You have to revise the paper in accordance with editor and reviewers comments. Please create an "author response" file with a point-by-point response to each comment, clearly describing how they have been addressed in the revision version.

We would appreciate receiving your revised manuscript by Feb 14 2020 11:59PM. To enhance the reproducibility of your results, we recommend that if applicable you deposit your laboratory protocols in protocols.io, where a protocol can be assigned its own identifier (DOI) such that it can be cited independently in the future. For instructions see: http://journals.plos.org/plosone/s/submission-guidelines#loc-laboratory-protocols

We look forward to receiving your revised manuscript.

Kind regards,

Aneta Agnieszka Koronowicz, PhD

Academic Editor

PLOS ONE

Journal Requirements:

Additional Editor Comments:

In the study was performed sensory analysis, but that no information is provided regarding participant consent (it is necessary for prospective studies, even in cases where the study has been deemed on “low risk” by an ethics committee). Please complete this information.

Reviewers' comments:

Reviewer's Responses to Questions

**Comments to the Author**

1. Is the manuscript technically sound, and do the data support the conclusions?

Reviewer #1: Yes

Reviewer #2: Yes

2. Has the statistical analysis been performed appropriately and rigorously? 

Reviewer #1: Yes

Reviewer #2: Yes

3. Have the authors made all data underlying the findings in their manuscript fully available?

Reviewer #1: Yes

Reviewer #2: Yes

4. Is the manuscript presented in an intelligible fashion and written in standard English?

Reviewer #1: Yes

Reviewer #2: Yes

5. Review Comments to the Author

Reviewer #1: Dear Authors,

your paper entitled: ‘Pulp obtained after isolation of starch from red and purple potatoes (Solanum tuberosum L.) as an innovative ingredient in the production of gluten free bread’ has been sent for my consideration. In it, you investigate the influence of different levels of potato pulp from red and purple potatoes on the content of polyphenols and fiber in gluten-free breads. It is a good, interesting work, however during the reading, the following remarks came to my mind:

General: I suggest replacing 'gluten free' with 'gluten-free'. The authors use both forms at work. This should be unified.

Lines 23-24: This is not entirely true. The use of whole-grain flour, e.g. from amaranth or quinoa, also provides bioactive compounds. Besides, are only antioxidants important compounds? Please, in the summary and in the introduction, approach the subject more broadly.

Keywords: Please avoid repeating the words used in the title.

Lines 39-41: How much potato pulp is produced during starch production? Please mention the second by-product, i.e. potato juice and (briefly) the possibilities of its use.

General: The authors point out in the introduction that the gluten-free diet is low in minerals, protein, etc. Then they add potato pulp to enrich the bread with fiber, which hinders the absorption of nutrients. Can the authors explain how increasing dietary fiber content can help in qualitative malnutrition?

Lines 105; 163; 169 etc.: ‘freeze dried’ -> ‘freeze-dried’

Line 106: total starch or maybe just one of them? Potato pulp contains much more water than starch. Did the replacement include water content?

Line 120: Please convert rpm to g or rcf

Section ‘Acrylamide content’: The authors correctly described the results obtained, however, please refer to food safety. What are the limits for AA?

Reviewer #2: The authors of the manuscript raise a very important, from the point of view of the circular economy, aspect of by-product management that arises after the isolation of starch from red and purple potatoes. Among other things, due to the fact that the remaining pulp is rich in health-related compounds, both in the nature of polyphenols and dietary fiber, a possible direction of its use could be its use as an innovative fraction of dietary fiber, for example in the production of gluten-free bread. The obtained results allow us to plant a successfully applied test of pulp application, whereas red potato pulp is characterized by better health potential.

In conclusion, the manuscript is interesting, zarówno ze wzgledu na aspekt poznawczy, jak I aplikacyjny. I recommend the manuscript to be published. I suggest minor editorial changes:

Line 3 - put the star symbol "*" next to Gumul - this person is the author of correspondence

Line 37 – it is suggested to add at least a literature reference in which the pulp production method would be described.

Line 120 – 4000 rpm – spin speed should be expressed as the value of centrifugal force "g"

Line 121 – reference should be given

Line 126,128,129,130,132 – the bibliography entry should be standardized

Line 143, 195 – Marconi et al. should be Matrconi et al.

W Tab. 1, 2, 3, 4 - g/100g d.m. should be g 100g-1 d.m.

significance levels (letters) should be placed in superscript

- there should be a dots after the titles of the tables and figures

- in table 4 – microgramy/kg should be µg kg-1

- in table 4 the acrylamide values should be given to the first decimal place

- Line 351 - at the end of a sentence put a dot.

6. PLOS authors have the option to publish the peer review history of their article (what does this mean?). If published, this will include your full peer review and any attached files.

Reviewer #1: No

Reviewer #2: No

---

## [Author Response · Author response to Decision Letter 0]

31 Jan 2020

Additional Editor Comments:

In the study was performed sensory analysis, but that no information is provided

regarding participant consent (it is necessary for prospective studies, even in

cases where the study has been deemed on “low risk” by an ethics

committee). Please complete this information.

According to the regulations in Poland, the consent of the bioethics committee is required only with respect to medical experiments (Regulation of the Minister of Health and Social Welfare on detailed rules for appointing and financing and the mode of operation of bioethics committees - Dz. U. 1999 No 47 item 480). In turn, the Act on the professions of a doctor and a dentist (Journal of Laws 2019, item 537 as amended - hereinafter: JoL) distinguishes two types of medical experiments - therapeutic and research. According to article 21 section 2 JoL "A medical experiment is the introduction by a doctor of new or only partially tested diagnostic, therapeutic or prophylactic methods ...". And "A research experiment is primarily aimed at extending medical knowledge ...". (Article 21(3) JoL). In addition, in accordance with the rules of the Pharmaceutical Law Act (Journal of Laws 2019, item 499, as amended), the tasks of the Bioethics Committee also include providing opinions on clinical trials of medicinal products (Article 37l, paragraph 1).

Gluten-free breads containing potato pulp are not a medicinal product but a food product. It follows from the above, that in the case of the studies presented in our article the consent of the bioethics committee was not required. Sensory evaluation is widely used in our studies on food products and the committee's consent was never required when publishing the results of such studies. Moreover, it does not appear in scientific publications of other authors known to us.

All the panelists who participated in sensory evaluation voluntarily agreed to be involved in the survey and were informed about the type of additive included in bread formulation. 

Example papers;

Jarosław Korus, Mariusz Witczak, Rafał Ziobro, Lesław Juszczak, Hemp (Cannabis sativa subsp sativa) flour and protein preparation as natural nutrients and structure forming agents in starch based gluten-free bread Lwt-Food Science and Technology, 2017, DOI: 10.1016/j.lwt.2017.05.046

Magdalena Krystyjan, Dorota Gumul, Rafał Ziobro, Marek Sikora, The Effect of Inulin as a Fat Replacement on Dough and Biscuit Properties, Journal of Food Quality, 2015, 38 (15), 305-315

Gumul D, Areczuk A., Ziobro R., Ivanisova E, Zieba T. The influence of freeze-dried red and purple potatoes on content of bioactive polyphenolic compounds and quality properties of extruded maize snacks. Quality Assurance and Safety of Crops and Foods. 2017, doi. 10.3920.QAS. 2016.1055.

 

Reviewer’s comments

Review Comments to the Author

Thank you very much for the comments. We have tried to correct the manuscript according to the suggestions. Please find below the answers for specific comments:

Reviewer #1: 

General: I suggest replacing ‘gluten free’ with ‘gluten-free’;. The authors use both forms at work. This should be unified.

applied

Lines 23-24: This is not entirely true. The use of whole-grain flour, e.g. from

amaranth or quinoa, also provides bioactive compounds. Besides, are only

antioxidants important compounds? Please, in the summary and in the introduction,

approach the subject more broadly.

Appropriate text was added to the abstract and the introduction. 

Keywords: Please avoid repeating the words used in the title.

Corrected according to the suggestion.

Lines 39-41: How much potato pulp is produced during starch production? Please

mention the second by-product, i.e. potato juice and (briefly) the possibilities of

its use.

Appropriate text was added to the introduction.

General: The authors point out in the introduction that the gluten-free diet is low

in minerals, protein, etc. Then they add potato pulp to enrich the bread with fiber,

which hinders the absorption of nutrients. Can the authors explain how increasing

dietary fiber content can help in qualitative malnutrition?

The content of minerals and other nutrients was increased by the applied addition of potato pulp. Appropriate text explaining this issue was added to the introduction.

Lines 105; 163; 169 etc.: ‘freeze dried’ -> ‘freeze-dried’

corrected

Line 106:

total starch or maybe just one of them? Potato pulp contains much more water than

starch. Did the replacement include water content?

corrected

Line 120: Please convert rpm to g or rcf

corrected

Section ‘Acrylamide content’ The authors correctly described the

results obtained, however, please refer to food safety. What are the limits for AA?

Appropriate text was added to the section. 

Reviewer #2

Line 3 - put the star symbol ‘*’ next to Gumul - this person is the author

of correspondence

corrected

Line 37 – it is suggested to add at least a literature reference in which the

pulp production method would be described.

Additional reference was added.

Line 120 – 4000 rpm – spin speed should be expressed as the value of

centrifugal force “g”

corrected

Line 121 – reference should be given

Additional reference was added.

Line 126,128,129,130,132 – the bibliography entry should be standardized

corrected

Line 143, 195 – Marconi et al. should be Matrconi et al.

corrected

Tab. 1, 2, 3, 4 - g/100g d.m. should be g 100g-1 d.m.

corrected

significance levels (letters) should be placed in superscript

- there should be a dots after the

corrected

titles of the tables and figures

- in table 4 – microgramy/kg should be µg kg-1

corrected

- in table 4 the acrylamide values should be given to the first decimal place

corrected

- Line 351 - at the end of a sentence put a dot.

corrected

---

## [Decision Letter · Decision Letter 1]

27 May 2020

PONE-D-19-31585R1

Pulp obtained after isolation of starch from red and purple potatoes (Solanum tuberosum L.)  as an innovative ingredient in the production of gluten-free bread

PLOS ONE

Dear Dr. Gumul,

Thank you for submitting your manuscript to PLOS ONE. After careful consideration, we feel that it has merit but does not fully meet PLOS ONE’s publication criteria as it currently stands. Therefore, we invite you to submit a revised version of the manuscript that addresses the points raised during the review process.

We look forward to receiving your revised manuscript.

Kind regards,

Juergen Koenig

Academic Editor

PLOS ONE

Reviewers' comments:

Reviewer's Responses to Questions

**Comments to the Author**

1. If the authors have adequately addressed your comments raised in a previous round of review and you feel that this manuscript is now acceptable for publication, you may indicate that here to bypass the “Comments to the Author” section, enter your conflict of interest statement in the “Confidential to Editor” section, and submit your "Accept" recommendation.

Reviewer #1: All comments have been addressed

Reviewer #2: All comments have been addressed

Reviewer #3: (No Response)

2. Is the manuscript technically sound, and do the data support the conclusions?

Reviewer #1: Yes

Reviewer #2: Yes

Reviewer #3: Yes

3. Has the statistical analysis been performed appropriately and rigorously? 

Reviewer #1: Yes

Reviewer #2: Yes

Reviewer #3: No

4. Have the authors made all data underlying the findings in their manuscript fully available?

Reviewer #1: (No Response)

Reviewer #2: Yes

Reviewer #3: Yes

5. Is the manuscript presented in an intelligible fashion and written in standard English?

Reviewer #1: (No Response)

Reviewer #2: (No Response)

Reviewer #3: Yes

6. Review Comments to the Author

Reviewer #1: The authors have tried a lot to improve the whole document. The comments from the rest of the reviewers helped a lot.

Considering the important improvements I believe the work has reached an acceptable level and it can be published on PLoS One.

Reviewer #2: (No Response)

Reviewer #3: (No Response)

7. PLOS authors have the option to publish the peer review history of their article (what does this mean?). If published, this will include your full peer review and any attached files.

Reviewer #1: No

Reviewer #2: No

Reviewer #3: No

---

## [Author Response · Author response to Decision Letter 1]

19 Jun 2020

The answers to prof. Berghofer's comments are submitted in the attached file (only the comments, the changes in the text are not included).

We would like to thank for the comments, as they substantially improved the quality of the paper. 

Best regards

Dorota Gumul and co-workers

---

## [Editor Report · Decision Letter 2]

6 Jul 2020

Pulp obtained after isolation of starch from red and purple potatoes (Solanum tuberosum L.)  as an innovative ingredient in the production of gluten-free bread

PONE-D-19-31585R2

Dear Dr. Gumul,

We’re pleased to inform you that your manuscript has been judged scientifically suitable for publication and will be formally accepted for publication once it meets all outstanding technical requirements.

Kind regards,

Juergen Koenig

Academic Editor

PLOS ONE
---

## [Editor Report · Acceptance letter]

24 Feb 2020

PONE-D-19-31585R1 

Pulp obtained after isolation of starch from red and purple potatoes (Solanum tuberosum L.)  as an innovative ingredient in the production of gluten-free bread 

Dear Dr. Gumul:

I am pleased to inform you that your manuscript has been deemed suitable for publication in PLOS ONE. Congratulations! Your manuscript is now with our production department. 

With kind regards,

on behalf of

Prof. Aneta Agnieszka Koronowicz 

Academic Editor

PLOS ONE